# Definition of High-Risk Early Hormone-Positive HER2−Negative Breast Cancer: A Consensus Review

**DOI:** 10.3390/cancers14081898

**Published:** 2022-04-09

**Authors:** Mattia Garutti, Gaia Griguolo, Andrea Botticelli, Giulia Buzzatti, Carmine De Angelis, Lorenzo Gerratana, Chiara Molinelli, Vincenzo Adamo, Giampaolo Bianchini, Laura Biganzoli, Giuseppe Curigliano, Michelino De Laurentiis, Alessandra Fabi, Antonio Frassoldati, Alessandra Gennari, Caterina Marchiò, Francesco Perrone, Giuseppe Viale, Claudio Zamagni, Alberto Zambelli, Lucia Del Mastro, Sabino De Placido, Valentina Guarneri, Paolo Marchetti, Fabio Puglisi

**Affiliations:** 1CRO Aviano, National Cancer Institute, IRCCS, 33081 Aviano, Italy; lorenzo.gerratana@uniud.it (L.G.); fabio.puglisi@cro.it (F.P.); 2Department of Surgery, Oncology and Gastroenterology, University of Padova, 35100 Padova, Italy; gaia.griguolo@unipd.it (G.G.); valentina.guarneri@unipd.it (V.G.); 3Division of Oncology 2, Istituto Oncologico Veneto IRCCS, 35100 Padova, Italy; 4Department of Radiological, Oncological and Pathological Sciences, Sapienza University of Rome, Policlinico Umberto I, 00100 Rome, Italy; andrea.botticelli@uniroma1.it; 5Department of Medical Oncology, IRCCS Ospedale Policlinico San Martino, 16100 Genova, Italy; giulia.buzzatti@gmail.com (G.B.); chiara.molinelli91@gmail.com (C.M.); lucia.delmastro@hsanmartino.it (L.D.M.); 6Department of Clinical Medicine and Surgery, University of Naples Federico II, 80100 Naples, Italy; carmine.deangelis1@unina.it (C.D.A.); deplacid@unina.it (S.D.P.); 7Department of Human Pathology, Papardo Hospital, University of Messina, 89121 Messina, Italy; vincenzo.adamo@unime.it; 8Department of Medical Oncology, IRCCS Ospedale San Raffaele, 20132 Milan, Italy; bianchini.giampaolo@hsr.it; 9School of Medicine and Surgery, Università Vita-Salute San Raffaele, 20020 Milan, Italy; 10Ospedale Santo Stefano, Prato Sandro Pitigliani Medical Oncology Division, Hospital of Prato, 59100 Prato, Italy; laura.biganzoli@uslcentro.toscana.it; 11Division of New Drugs and Early Drug Development, European Institute of Oncology IRCCS, 20100 Milan, Italy; giuseppe.curigliano@ieo.it; 12Department of Oncology and Hemato-Oncology, University of Milan, 20122 Milan, Italy; giuseppe.viale@ieo.it; 13Department of Breast and Thoracic Oncology, IRCCS INT Fondazione G. Pascale, 80144 Napoli, Italy; m.delaurentiis@breastunit.org; 14Precision Medicine in Breast Cancer Unit, Department of Woman and Child Health and Public Health, IRCCS, Scientific Directorate, Fondazione Policlinico Universitario A. Gemelli, 00168 Rome, Italy; alessandra.fabi@policlinicogemelli.it; 15Department of Traslational Medicine and for Romagna, Clinical Oncology, S Anna University Hospital, Università degli Studi di Ferrara, 44121 Ferrara, Italy; a.frassoldati@ospfe.it; 16Department of Translational Medicine, Università del Piemonte Orientale, 28100 Novara, Italy; alessandra.gennari@uniupo.it; 17Azienda Ospedaliero-Universitaria Maggiore della Carità, 28100 Novara, Italy; 18Candiolo Cancer Institute, FPO IRCCS, 10060 Candiolo, Italy; caterina.marchio@unito.it; 19Department of Medical Sciences, University of Turin, 10126 Turin, Italy; 20Clinical Trials Unit, Istituto Nazionale Tumori di Napoli, IRCCS Fondazione Pascale, 80144 Naples, Italy; f.perrone@istitutotumori.na.it; 21Department of Pathology, European Institute of Oncology IRCCS, 20122 Milan, Italy; 22Medical Oncology Unit, Istituto di Ricovero e Cura a Carattere Scientifico (IRCCS), Azienda Ospedaliero-Universitaria di Bologna, 40100 Bologna, Italy; claudio.zamagni@aosp.bo.it; 23Breast Cancer Section Department of Biomedical Sciences, IRCCS Humanitas Research Hospital, Humanitas University, Rozzano, 20089 Milan, Italy; alberto.zambelli@hunimed.eu; 24Dipartimento di Medicina Interna e Specialità Mediche, University of Genova, 16159 Genova, Italy; 25IRCCS Istituto Dermopatico dell’Immacolata (IDI-IRCCS), 00167 Rome, Italy; paolo.marchetti@uniroma1.it; 26Department of Medicine, University of Udine, 33100 Udine, Italy

**Keywords:** breast cancer, hormone receptors, adjuvant, endocrine therapy, chemotherapy, risk of relapse, genomic signature, ctDNA, TNM, consensus

## Abstract

**Simple Summary:**

Breast cancer is one of the major causes of cancer-related morbidity and mortality in women worldwide. Despite recent improvements in adjuvant treatment of hormone receptor-positive/HER2−negative breast cancer, estimating the risk of relapse of breast cancer on an individual basis is still challenging. The IRIDE (high risk definition in breast cancer) working group was established with the aim of reviewing evidence from the literature to synthesize the current relevant features that predict hormone-positive/HER2−negative early breast cancer relapse. This work may guide clinicians in the practical management of hormone-positive/HER2−negative early breast cancers.

**Abstract:**

Breast cancer is one of the major causes of cancer-related morbidity and mortality in women worldwide. During the past three decades, several improvements in the adjuvant treatment of hormone receptor-positive/HER2−negative breast cancer have been achieved with the introduction of optimized adjuvant chemotherapy and endocrine treatment. However, estimating the risk of relapse of breast cancer on an individual basis is still challenging. The IRIDE (hIGh Risk DEfinition in breast cancer) working group was established with the aim of reviewing evidence from the literature to synthesize the current relevant features that predict hormone-positive/HER2−negative early breast cancer relapse. A panel of experts in breast cancer was involved in identifying clinical, pathological, morphological, and genetic factors. A RAND consensus method was used to define the relevance of each risk factor. Among the 21 features included, 12 were considered relevant risk factors for relapse. For each of these, we provided a consensus statement and relevant comments on the supporting scientific evidence. This work may guide clinicians in the practical management of hormone-positive/HER2−negative early breast cancers.

## 1. Introduction

Breast cancer (BC) is one of the major causes of cancer-related morbidity and mortality in women worldwide [1].

BC is generally subdivided into four distinct subtypes according to the expression of hormone receptors (HR; estrogen receptor (ER) and progesterone receptor (PgR)) and human epidermal growth factor receptor-2 (HER2): HR+/HER2−, HR+/HER2+, HR/HER2+ and triple-negative BC. HR+/HER2− BC represents the most common subtype, accounting for around 70% of all BC cases. Moreover, the majority (more than 90%) of HR+/HER2− BC primary diagnoses occur in non-metastatic stages (stages I–III) [2].

During the past three decades, several improvements in the adjuvant treatment of HR+/HER2− BC have been achieved with the introduction of optimized chemotherapy regimens (e.g., sequential anthracycline and taxane-based chemotherapy and dose-dense regimens) and optimized adjuvant endocrine treatment (e.g., aromatase inhibitors, extended endocrine therapy, and, more recently, ovarian suppression for the treatment of premenopausal patients) [3,4,5,6]. Moreover, novel targeted agents, such as cyclin-dependent kinase 4 and 6 (CDK4 and 6) inhibitors or poly(ADP-ribose) polymerase (PARP) inhibitors for germline *BRCA*-mutated patients, might become available in the near future for patients at high risk of recurrence despite the standard adjuvant treatment [7,8]. 

In this context, adequately estimating the risk of BC relapse on an individual basis is essential. On one hand, unmet needs and the potential for the implementation of new treatments exist for those at higher risk of recurrence, whereas on the other hand, there is a significant risk of overtreatment for patients at a lower risk of relapse.

A multitude of putative prognostic factors associated with risk of relapse (distant and loco-regional) have been described in HR+/HER2− BC; however, the level of evidence and the relevance of each of these factors varies significantly. In addition to the canonical well-consolidated prognostic factors, such as nodal involvement and tumor size, for several novel molecular prognostic factors, such as gene-expression signatures, consistent data have been generated leading to implementation in clinical practice, while other putative prognostic factors (e.g., evaluation of circulating tumoral DNA (ctDNA)) are currently in an advanced stage of investigation. Moreover, in clinical practice prognostic information derived from each factor is never considered separately, but combined to obtain a more accurate risk estimate [9,10,11]. In addition, both therapeutic options for early HR+/HER2− BC and knowledge regarding clinical and biological factors are continuously evolving. Therefore, an up to date revision of the available evidence concerning factors that guide adjuvant treatment in early BC is needed.

However, in this complex and rapidly changing situation, not all clinical scenarios can be directly informed by data from randomized trials. Therefore, the present consensus panel was established with the rationale of identifying the clinical, pathological, morphological, and genetic factors potentially related to a high risk of recurrence, in order to better inform treatment decisions in patients with HR+/HER2− early BC (eBC). In the context of the recent increase in potential treatment options for patients with HR+/HER2− eBC, and of evidence concerning the use of gene-expression assays in HR+/HER2− eBC, a comprehensive and updated view of the available data is needed in order to identify the clinically significant recurrence risk factors and to better inform day-to-day clinical decisions.

## 2. Materials and Methods

A panel of 25 experts in BC was involved in identifying the clinical, pathological, morphological, and genetic factors potentially related to a high risk of recurrence in patients with HR+/HER2− eBC (the factors are summarized in Table 1). A scoping literature search was performed for each factor. The databases used were PUBMED and Web of Science. In addition, the reference lists of the included articles were searched for additional articles of interest. The articles included original papers, randomized controlled trials, systematic reviews or meta-analyses, guidelines, and consensus statements. Articles and case reports that were not written in English were excluded. The search was conducted in April 2021. The relevant papers were ranked, and the level of overall evidence graded according to the United States Preventative Services Task Force (USPFTF) as low, moderate, and good.

Based on the results of the literature review and their personal opinion, the panel participated in two rounds of an online RAND [12] survey (the first in June 2021 and the second in July 2021) to define the relevance in relation to the prognostic value of each factor. A nine-point scale was used to quantify relevance in relation to the prognostic value, 1 was considered “not relevant”, 2–3 “poorly relevant”, 4–6 “moderately relevant”, and 7–9 “relevant”. The median score was used to classify relevance, and the 30th–70th interpercentile range corrected for asymmetry was used to assess disagreement. After viewing the results of the first round, in which their responses were highlighted, panel members were asked to review their choices in the second round. Results of the second round are shown in Figure 1.

The final meeting was held online on 11 October 2021. The goal was to examine each factor that was defined as relevant in the RAND survey, and to reach a consensus on the final statements describing specific characteristics of each factor.

## 3. Results

The features considered highly relevant by the panelists for the recurrence of HR+/HER2− surgically removed stage I–III eBC are presented here. Each feature will be characterized by consensus statements and evidence from the literature. 

### 3.1. Histological Grade

#### 3.1.1. Consensus Statements 

The histological grade is a feature associated with the aggressiveness of tumoral lesions and risk of relapse of surgically resected HR+/HER2− eBC;High histological grade (Grade 3) is characterized by a poor prognosis.

#### 3.1.2. Source of Evidence

Histological grade, evaluated using the Nottingham scoring system, which takes into account tubule formation, nuclear pleomorphism, and mitotic activity, has been identified as a relevant prognostic factor in patients diagnosed with HR+/HER2− eBC in several randomized trials and large case series [13,14,15,16,17,18,19,20].

Even if in some trials (e.g., BIG 1-98 trial) a progressive worsening of patient prognosis has been observed with the increase in histological grade [20], other studies did not identify significant differences in prognosis between patients with well-differentiated (Grade 1) and moderately differentiated (Grade 2) tumors at multivariate analysis [17]. These discrepancies might, at least in part, be explained by issues in the evaluation and interpretation of intermediate histological grade. A certain degree of inconsistency in the evaluation of histological grade has been described when the same sample was evaluated by different pathologists [21]. Moreover, when assessed using gene-expression signatures, moderately differentiated (Grade 2) samples can be reassigned to either well-differentiated (Grade 1) or poorly differentiated (Grade 3) subgroups [22].

Despite these issues, histological grade has been confirmed to be a clinically relevant prognostic factor associated with prognosis, both at short- (2 years [20]) and long-term (20 years [16]) follow-up. An Early Breast Cancer Trialists Collaborative Group (EBCTCG) meta-analysis, including more than 13,000 patients with HR+ eBC treated with adjuvant endocrine treatment, showed a progressive increase in risk of distant relapse at 20 years with the increase in histological grade (10%, 13% and 17% for Grade 1, Grade 2 and Grade 3, respectively) [16].

The histological grade evaluated by the Nottingham scoring system is, therefore, a highly consolidated prognostic factor and has recently been incorporated in BC staging, according to the 8th edition of the American Joint Committee on Cancer (AJCC) BC staging system [17,18,23,24].

### 3.2. Histological Type

#### 3.2.1. Consensus Statements

The histological type is a feature correlated to the risk of relapse of surgically resected HR+/HER2− eBC;Pure tubular, pure mucinous and pure cribriform histologies are characterized by a good prognosis.

#### 3.2.2. Source of Evidence

The histological type equates to a descriptive classification of the morphological and spatial growth pattern of cancer cells. The association of some histological types with eBC prognosis has been observed. In particular, starting from the College of American Pathologists (CAP) consensus statement of 1999, the histological type has been included in the category of prognostic factors in all international BC guidelines [17,25,26,27].

According to the 2019 World Health Organization classification of BC [28], among HR+ BC the tubular, mucinous, and cribriform histologies are characterized by a good prognosis, and some guidelines support a de-escalation strategy for these special histologies [25,26].

Pure tubular BCs (tBCs) represent 1–2% of all BCs and are microscopically characterized by well-differentiated tubular structures [29]. From a molecular perspective, tBCs often present chromosome 16q loss (78–86%), chromosome 1q gain (50–62%) and have a luminal A gene expression profile [30]. Several studies suggest a favorable post-surgical prognosis with a 10-year disease-free survival (DFS) of up to 99.1% and a 10-year overall survival (OS) of up to 100% [30,31]. The local or systemic relapse rate is less than 8% [32]. 

Pure mucinous BCs (muBCs) account for nearly 2% of BCs and are microscopically characterized by abundant production of mucin [33]; a molecular luminal A pattern has frequently been observed in muBCs [34]. Generally, muBCs have a good post-surgical prognosis, with a 5-year DFS of up to 94% and a 20-year OS of 81% [35,36]. Moreover, a multivariate analysis of 11,422 patients has shown that nodal involvement might be a relevant prognostic factor, even more than tumor size, since most of the tumor volume is composed of mucin [35,37].

Pure cribriform BCs (cBCs) represent <4% of BCs and are microscopically defined as atypical cells organized in a sieve-like pattern surrounded by a fibrosclerotic stroma [38]. From a molecular perspective, cBCs mainly belong to the luminal A class [38]. Clinically, cBCs are characterized by a favorable post-surgical prognosis with a 10-year OS of up to 100% [38].

Comprehensively, the available evidence, although mostly in the form of retrospective studies, suggests that tBCs, muBCs, and cBCs represent favorable prognostic histologies in eBC. It is important to highlight that to qualify for a “pure” special type, 90% or more of the neoplastic cell population should belong to the relevant histological type [28].

### 3.3. Nodal Status (N)

#### 3.3.1. Consensus Statements

Nodal status is a feature correlated with the risk of relapse of surgically resected HR+/HER2− eBC;Nodal involvement is a well-known prognostic factor in HR+/HER2− BC and is associated with worse prognosis both in the short and long term. Patient prognosis worsens progressively with the increased number of involved axillary lymph nodes.

#### 3.3.2. Source of Evidence

Nodal involvement in clinical and pathological examination is one of the best known negative prognostic factors for all BC subtypes, including HR+/HER2− eBC, as reported by several randomized clinical trials and meta-analyses [16,17,18,20,39,40,41,42,43,44]. In the BIG 1-98 trial, which included 7707 patients, nodal involvement was significantly associated with risk of relapse in the first years after diagnosis [20]. Moreover, a meta-analysis by Pan et al. identified nodal status as a very strong risk factor for distant BC relapse over the period of 5–20 years after diagnosis. For example, in this meta-analysis, among patients with T1 disease, the risk of distant recurrence was 13% if no nodal involvement was observed (pT1N0), 20% if one to three involved nodes were identified (pT1pN1a), and 34% for patients with four to nine nodes involved (pT1pN2a). Among patients with T2 disease, the risk of distant recurrence was 19% with pT2pN0, 26% with pT2pN1a, and 41% with pT2pN2a [16]. Even among patients who achieve a pathological complete response (pCR) after neoadjuvant treatment, the presence of nodal involvement at the initial clinical staging remains a significant risk factor for relapse [40]. 

Consistently, nodal status is included in the AJCC BC staging system [17,18].

It should be noted, however, that the current surgical strategy of avoiding axillary dissection in selected cases, despite the occurrence of metastases in one or two sentinel lymph nodes, may lead to the downstaging of the tumor in quite a large number of patients. 

### 3.4. Tumor Size (T)

#### 3.4.1. Consensus Statements

Primary tumor size is a feature correlated to the risk of relapse in surgically resected HR+/HER2− eBC;Primary tumor size is a well-known prognostic factor in HR+/HER2− BC; an increase in primary tumor size is associated with worse prognosis both in the short and long term.

#### 3.4.2. Source of Evidence

Primary tumor size is a well-known prognostic factor in HR+/HER2− BC and is included in the AJCC BC staging system [16,17,18,20].

In the BIG 1-98 trial, which included 7707 patients, a primary tumor size above 2 cm was reported to be significantly associated with risk of relapse in the early years after diagnosis [20]; however, in a population registry including 3844 women treated with adjuvant tamoxifen, primary tumor size was not significantly associated with short-term risk of relapse [45]. Moreover, a recent meta-analysis by Pan et al. identified primary tumor size as a significant negative prognostic factor for distant BC relapse over the period of 5–20 years after diagnosis. For example, in this meta-analysis, among patients with T1 disease, the risk of distant recurrence varied from 13% to 34% according to lymph node involvement, whereas it varied from 19% to 41% among patients with T2 disease [16]. In addition, a recently published study that included 1685 patients, reported a two-fold increase in risk of BC death for patients with primary tumors >5 cm (T3/T4) [44].

### 3.5. Ki-67

#### 3.5.1. Consensus Statements

The Ki-67 labeling index (L.I.) is a feature correlated to the risk of relapse of surgically resected HR+/HER2− eBC and its prognostic role is well consolidated;A high Ki-67 L.I. is associated with the worst prognosis; the cut-off for low risk of relapse is <20% and that for high risk of relapse is >30%.

#### 3.5.2. Source of Evidence

Ki-67 is a protein encoded by the *MKI67* gene that is expressed in all but G0 cell cycle phases (i.e., G1, S, G2, and M) and represents a robust biomarker of cellular proliferation [46]. Clinically, Ki-67 expression levels are assessed by immunohistochemistry [46].

Ki-67 was proposed as a prognostic biomarker for eBC, although a questionable analytical validity has, somehow, undermined its use in the clinical decision-making process [46,47]. However, in a huge international effort, many of the crucial technical problems of Ki-67 evaluation have been addressed and its prognostic role in HR+/HER2− eBC has been reinforced [47]. Several lines of evidence have shown that higher Ki-67 L.I. are associated with the poorest prognosis [48,49]. Nevertheless, an unequivocal and universal shared threshold to define the high or low value of Ki-67 is still debated. For example, the International Ki-67 Working Group stated that a cut-off of ≤5% can be used to define a good prognosis group among HR+/HER2+ eBC whereas a level of ≥30% is associated with the poor prognosis group [47]. In the European Society of Medical Oncology (ESMO) eBC guidelines, the Ki-67 cut-off is defined as per local laboratory value (for example, if the laboratory median score is 20%, the value of ≥30% can be considered clearly high whereas those of ≤10% can be considered clearly low) [26]. Moreover, two different meta-analyses with a total of 4500 and 64,000 patients have identified a cut-off for Ki-67 of 19% and 25%, respectively [50,51]. Again, a high Ki-67 L.I. is associated with reduced DFS and OS [50,51]. 

Comprehensively, the available evidence shows that high and low Ki-67 L.I. are associated with a poor and good prognosis, respectively. Although a universally shared cut-off is still debated, it appears reasonable to consider a value >30% as high and a value <20% as low.

### 3.6. Expression Level of Hormonal Receptors (Estrogen Receptor, Progesterone Receptor)

#### 3.6.1. Consensus Statements

Expression level of HRs (ER, PgR) is a feature correlated to the risk of relapse in surgically resected HR+/HER2− eBC;Expression level of HRs is a well-known prognostic factor and a predictive factor for response to endocrine treatment in surgically resected HR+/HER2− BC;A low level of expression of ERs is associated with a high risk of relapse, whereas high levels of expression are associated with a better prognosis in relation to a higher sensitivity to endocrine treatment;Absent or low (<20%) expression of PgR, even in the presence of ER expression, represents a negative prognostic factor, potentially indicative of uncertain endocrine sensitivity.

#### 3.6.2. Source of Evidence

ER and PgR expression is evaluated using immunohistochemistry and is reported as the percentage of positive tumor nuclei. According to the current American Society of Clinical Oncology (ASCO)/CAP guidelines, the BC samples with ≥1% of positive tumor nuclei should be interpreted as ER+ [52]. However, the same guidelines suggest that samples with 1–10% of cells staining ER+ should be reported as “ER Low Positive” (2–3% of ER+ BCs). This rare subset of BC, despite being formally categorized as ER+ by ASCO-CAP guidelines, carries several clinical and biological analogies with triple-negative BC [52,53]. 

Expression levels of HRs carry important prognostic and predictive information regarding sensitivity to endocrine treatment, which are clinically relevant to guide therapeutic decisions. A significant association between higher levels of HR expression, as assessed by immunohistochemistry, and the benefit of endocrine treatment has been reported, even if several other biological factors can influence endocrine sensitivity (HER2 status, grading, Ki67).

In particular, higher levels of ER expression have been positively associated with OS and DFS [54]. Moreover, higher levels of ER expression are also associated with a greater sensitivity to endocrine treatment. A large meta-analysis, including 20 clinical trials, recently reported on the benefit of 5 years of tamoxifen according to ER and PgR levels as measured by the ligand-binding assay. For women with tumors having <10 fmol ER/mg protein, no evidence of benefit was apparent, whereas for those with BCs with low levels of ER (10–20 fmol ER/mg protein) likelihood of recurrence was reduced by one-third by the addition of 5 years of tamoxifen (rate ratio (RR) 0.67). Benefit increased with higher ER levels, even if the proportional effect at the highest ER levels (>200 fmol/mg) was only slightly better than that at weak ER levels (RR 0.52) [39]. 

PgR expression levels have been reported to be positively associated with OS and DFS [54]. Several studies have suggested that PgR carries independent prognostic information in addition to ER, in particular in premenopausal BC patients [55,56]. Moreover, a recent EBCTCG meta-analysis including more than 62,000 HR+/HER2− eBC patients treated with adjuvant endocrine treatment confirmed that PgR status represents an independent prognostic factor in the first 5 years after BC diagnosis [16]. On the other hand, the role of PgR expression levels in predicting benefit from endocrine treatment appears limited, as several studies have reported a marginal impact of PgR levels compared with ER levels [39,57].

For this reason, only ER is used as a predictor of benefit from adjuvant endocrine therapy; PgR levels add prognostic information to help stratify outcomes in the ER+ population. 

Consistently with its well-known prognostic role, HR status has been included in the AJCC BC staging system [17,18].

### 3.7. Residual Cancer Burden

#### 3.7.1. Consensus Statements

Residual cancer burden (RCB) after neoadjuvant treatment is a feature correlated to the risk of relapse in surgically resected HR+/HER2− eBC;A high RCB is associated with a worse prognosis and RCB-III identifies patients at the highest risk of relapse.

#### 3.7.2. Source of Evidence

Achievement of pCR is an established surrogate endpoint for long-term outcome in BC patients treated with neoadjuvant chemotherapy [58,59]. However, the simple dichotomization in pCR versus non-pCR is prognostically suboptimal, as some patients achieving pCR will relapse and some patients with residual disease will still have an excellent prognosis. Moreover, even if HR+/HER2− BC patients generally present lower pCR rates than triple-negative and HER2+ subtypes, the presence of residual disease after neoadjuvant chemotherapy does not necessarily translate to a poor outcome in this BC subtype [58].

RCB has been proposed as a more detailed evaluation of residual disease after neo-adjuvant treatment, taking into account relevant pathological characteristics with independent prognostic impact (bidimensional measurements of primary tumor bed area, overall cancer cellularity, percentage of cancer that is in situ disease, number of positive lymph nodes, and size of the largest node metastasis).

Since its proposal by Symmans et al. in 2007 [60], RCB has been validated as a strong predictor of long-term outcome beyond pCR in BC patients treated with neoadjuvant chemotherapy [61]. Moreover, RCB has been reported to be capable of better stratifying patients in each BC subtype separately, including HR+/HER2− BC [61,62]. In a recent combined analysis of the I-SPY2 trial, event-free survival worsened significantly per unit of RCB in every BC subtype (hazard ratio 1.75; 95% confidence interval, 1.45–2.16 for HR+/HER2− BC) [61].

Considering its solid clinical relevance, the evaluation of RCB has been included in the Breast International Group/North American Breast Cancer Group (BIG-NABCG) recommendations quantification of residual disease in clinical trials after neoadjuvant treatment [63].

### 3.8. Genomic Signatures

Genomic signatures (GSs) are molecular assays that analyze the level of expression of several genes by using RNA extracted from formalin-fixed paraffin-embedded tumor samples and provide a prognostic stratification of tumors (e.g., quantification of the risk of relapse/metastases at 10 years) [64]. However, to increase the prognostic yield, GSs also take into account some clinical features (see below). The GSs with strong levels of evidence are Oncotype DX^®^, MammaPrint^®^ (MP), Prosigna^®^, and Endopredict^®^ (EP) [64]. However, only Oncotype DX^®^ and MP have the highest level of evidence (1A).

It is important to highlight that each GS includes a unique set of clinical and molecular features, making the results of each GS intrinsically different and not overlapping. This means that patients classified as high-risk by a GS might have a different class of risk with another GS.

#### 3.8.1. Oncotype DX Breast Recurrence Score Test^®^

##### Consensus Statements

The Oncotype DX Breast Recurrence Score Test^®^ (ODX) value is a feature correlated to the risk of relapse of surgically resected HR+/HER2− eBC;ODX can identify patients with low, intermediate, and high risk of relapse.

##### Source of Evidence

ODX evaluates the risk of BC relapse at 10 years and the benefit of chemotherapy in addition to endocrine treatment [64]. ODX has been validated both retrospectively [65,66] and prospectively [67,68,69,70,71,72] and its level of evidence is 1A (ESMO [26]) and 1 (National Comprehensive Cancer Network (NCCN) [25]).

ODX evaluates, on a formalin-fixed paraffin-embedded sample, the level of expression of 21 genes to produce a recurrence score (RS) that is a continuous variable spanning from 0 to 100, where 0 corresponds to the lower risk and 100 to the higher risk of relapse [64,73]. To maximize its clinical applicability, three different groups of risk of relapse have been identified: the low-, intermediate-, and high-risk group [64]. Moreover, the intermediate-risk group has been enriched by clinical information (patient age, tumor grade, and diameter) to better identify who could benefit from the addition of chemotherapy [67,68,69,70,71,72].

The low-risk group has been defined as RS 0–10 or 0–17, depending on the study [67,68,69,70,71,72]. Consistent with data derived from the four main studies including more than 50,000 patients, this group has an excellent prognosis with only endocrine therapy. For example, it has been shown that the 5-year distant recurrence-free survival (DRFS) rate was >99%, the 5-year BC-specific survival (BCSS) rate was >99%, and the 5-year DFS rate was 94% [67,68,69,70,71,72]. Although these results apply mainly to N0 disease, some patients with N1 involvement could also have a favorable prognosis. In the RxPONDER trial, patients with T1–3 and N1 tumors with RS 0–25 had a 5-year invasive disease-free survival (IDFS) of 91.0% and the addition of chemotherapy did not increase this outcome [71]. However, in the subgroup analysis, it seems that premenopausal patients might still derive some benefits from the addition of chemotherapy to endocrine treatment [71].

The high-risk group has been defined as RS 26–100 or 31–100, depending on the study. Even when treated with both chemotherapy and endocrine therapy, N0 patients had an IDFS at 9 years of 67.9–85.7% and a DRFS at 9 years of 80.2–93.8% [67,68,69,70,71,72]. However, in N1 disease, the DFS at 10 years was only 43% with endocrine therapy and 55% with chemo-endocrine treatment [74,75].

The intermediate group was composed of patients not included in the low- and high-risk groups [67,68,69,70,71,72]; in the TAILORx trial, the intermediate group was defined by an RS of 11–25. Therefore, the prognosis of this group would be somewhere intermediate between the other two. However, this group seems to have been composed of a heterogeneous population. Although N0 patients aged ≥50 years seemed to derive no benefit from the addition of chemotherapy to endocrine therapy, N0 patient <50 years could have a better IDFS and/or DRFS with treatment escalation. However, this seems to be relevant only in the group with RS 21–25 and in the group with RS 16–20 and a clinical high-risk disease [69,70]. In N1 disease, only premenopausal patients could derive a benefit from the addition of chemotherapy [71] possibly because of the endocrine effect of chemotherapy in this younger population of patient.

#### 3.8.2. Mammaprint^®^

##### Consensus Statements

MP value is a feature correlated to the risk of relapse of surgically resected HR+/HER2− eBC;MP can identify patients with low and high risk of relapse.

##### Source of Evidence

MP evaluates the risk of BC relapse at 10 years [64]. MP has been validated both retrospectively [75,76,77] and prospectively [78,79,80] and its level of evidence is 1A (ESMO [26]) and 1 (NCCN [25]).

The MP evaluates, on a formalin-fixed paraffin-embedded sample, the level of expression of 70 genes to dichotomize patients with low or high risk of relapse [81]. For example, in the TRANSBIG Consortium validation trials [77], patients were labeled as high genomic risk (gHigh) if they had a 5-year distant metastasis-free survival (DMFS) <90%, whereas low genomic risk (gLow) patients have a 5-year DMFS >90%. However, to increase the clinical applicability of such an approach, MP has been combined with a modified version of the Adjuvant! Online tool, which is a calculator of BC relapse risk that is based on clinical data. Such a modified tool can dichotomize patients in low clinical risk (cLow) or high clinical risk (cHigh) classes by the tumor diameter, grade, and lymph node involvement (http://www.adjuvantonline.com (accessed on 14 February 2022), currently not accessible, table form can be found as supplementary material in the MINDACT publication [78]). The combination of MP and the Adjuvant! Online tool allows the identification of four groups of patients with HR+/HER2− and N0–1 BC characterized by a different DMFS risk.

These four categories were prospectively evaluated in the MINDACT trial [78,79,80]. In particular, in the cLow/gLow group the 8-year DMFS was favorable (94.7%), even in the absence of chemotherapy, whereas in the cHigh/gHigh group it was worse (85.9%) even after the administration of chemotherapy. As expected, the cLow/gHigh and cHigh/gLow groups had an intermediate 8-year DMFS (91.1% and 90.8%, respectively). The study showed that patients with cHigh but gLow disease have a good prognosis with only endocrine therapy, thus allowing chemotherapy to be spared in >40% of the patients with cHigh. However, with MP, premenopausal women with cHigh/gLow disease may also take advantage of adding chemotherapy to endocrine treatment.

#### 3.8.3. Prosigna^®^

##### Consensus Statements

Prosigna value is a feature correlated to the risk of relapse of surgically resected HR+/HER2− eBC;Prosigna risk of recurrence score combined with clinical data can identify three groups of patients characterized by a low, intermediate, and high risk of BC relapse.

##### Source of Evidence

Prosigna evaluates the risk of BC relapse at 10 years [64]. This test also provides the molecular intrinsic subtype of the tumor (i.e., molecular classification of BC into luminal A/B, HER2−enriched, basal-like subtypes [82]). Prosigna has been validated retrospectively in post-menopausal patients [75,83,84,85] and its level of evidence is 1B (ESMO [26]) and 2A (NCCN [25]).

Prosigna is a 50-gene signature (PAM50) performed on a formalin-fixed paraffin-embedded sample. The assay can be run in local laboratories equipped with the dedicated instrumentation. PAM50 is used in conjunction with tumor diameter to generate a continuous risk of BC recurrence (ROR) [64]. To increase the clinical usability, in N0 patients, the ROR score has been categorized into three groups characterized by a different 10-year DMFS [64,75,83,84,85]. The 10-year DMFS for the low-risk group (score 0–40) is >90%, for the intermediate group (score 41–60) 80–90% and for the high-risk group (score 61–100) <80%. In N1 patients, the ROR score can be categorized into two groups (0–40 low risk, 41–100 high risk). The cut-off of 10-year DMFS between these groups is nearly 80%.

#### 3.8.4. Endopredict^®^

##### Consensus Statements

The EP value is a feature correlated to the risk of relapse of surgically resected HR+/HER2− eBC;EP GS combined with clinical data (EPclin) can dichotomize BC patients treated with endocrine therapy in two groups characterized by a low and high risk of relapse.

##### Source of Evidence

EP evaluates the risk of BC relapse at 10 years [64]. EP has been validated retrospectively [75,86,87,88] and its level of evidence is 1B (ESMO [26]) and 2A (NCCN [25]).

EP is a 12−gene signature performed on a formalin-fixed paraffin-embedded sample that generates a score of risk of relapse [64]. The assay can be run in local laboratories equipped with the dedicated instrumentation. The EP score has been combined with clinical information (tumor size and lymph node involvement) to generate the EPclin score. EPclin has been dichotomized to obtain two different risk groups: EPclin low risk (score <3.3) and EPclin high risk (score ≥3.3) characterized by different prognoses [86,87,88].

A key publication, based on combined clinical data from the ABCSG-6 and ABCSG-8 clinical trials, demonstrated that the EPclin low group had a 10-year DRFS of 93.5%, whereas in the EPclin high it was 74.1% [86,88]. These data were further validated through the clinical data of the ATAC trial [75]. In particular, it has been shown that the Epclin low group had a 10-year DRFS of 94.2%, whereas in the EPclin high group it was 71.2%. Notably, both the ABCSG6/8 and ATAC trials were composed of patients treated with only endocrine therapy.

## 4. Conclusions

The IRIDE working group identified an updated list of relapse risk factors for HR+/HER2− eBC (Table 2). The accurate identification and reporting of these features is of pivotal importance in clinical practice, in order to personalize adjuvant treatment intensity. Although each of these factors could modulate the overall risk of relapse, to date a prediction tool that includes and weights each of them is still lacking. However, it appears reasonable to use the identified features to tailor the adjuvant therapy. A possible useful implication of this work resides in a well-characterized list of prognostic items that could represent an essential information list to include in daily clinical practice and clinical trials.

For most of the risk factors considered relevant in our analysis, there was a high agreement rate (Figure 1). Conversely, the agreement for the non-relevant features was generally low, underlying a substantial inconsistency of scientific literature. However, although ctDNA was considered only moderately relevant, a growing prognostic role for this factor might be expected in the near future.

Our comprehensive work produced by an expert consensus represents a pragmatic reference to help clinicians better stratify eBC patients for tailored adjuvant treatments. 

## Figures and Tables

**Figure 1 cancers-14-01898-f001:**
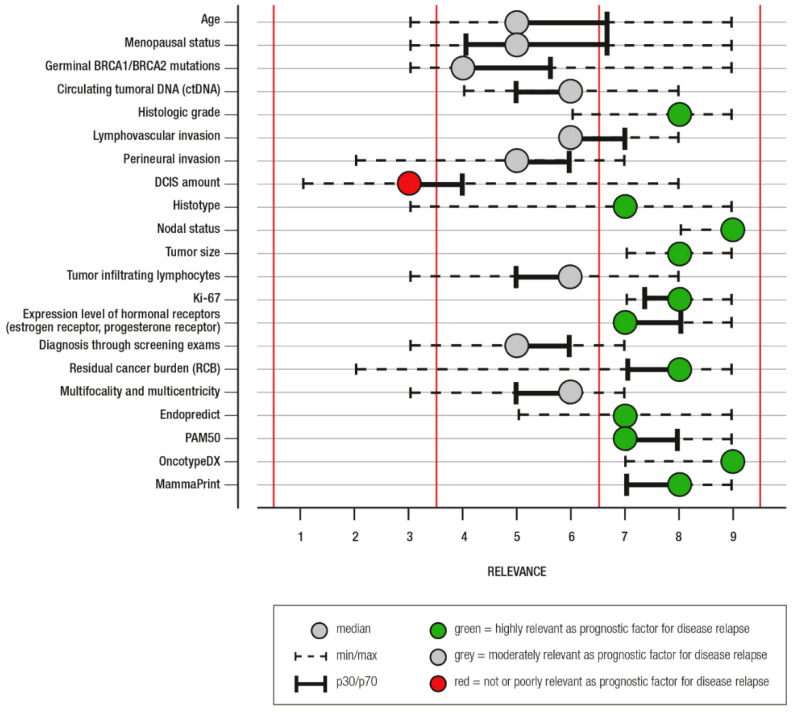
RAND results (2nd round): 1 = not relevant as prognostic factor for disease relapse; 2–3 = poorly relevant as prognostic factor for disease relapse; 4–6 = moderately relevant as prognostic factor for disease relapse; 7–9 = highly relevant as prognostic factor for disease relapse.

**Table 1 cancers-14-01898-t001:** Preliminary list of prognostic factors prognostic of for disease relapse.

**Clinical, Pathological, and Morphological Factors**
Age Menopausal status Germinal *BRCA1*/*BRCA2* mutations Circulating tumoral DNA Histological grade Lymphovascular invasion Perineural invasion DCIS amount Histotype Nodal status (N) Tumor size (T) Tumor-infiltrating lymphocytes Ki-67 Expression level of hormonal receptors (ER, PgR) Diagnosis through screening exams Residual cancer burden Multifocality and multicentricity
**Genomic Factors**
EndoPredict^®^ PAM50 Oncotype DX MammaPrint^®^

DCIS: ductal carcinoma in situ, ER: estrogen receptor, PgR: progesterone receptor.

**Table 2 cancers-14-01898-t002:** Synoptic table of risk factors for disease relapse.

Factor	High Risk	Low Risk
Grade	3	1
Histotype	n/a	Pure tubular, pure mucinous, pure cribriform
Tumor size	T3/4	T1
Nodal status	N2/N3	N0
Ki-67	>30%	<20%
Expression level of hormonal receptors (ER, PgR)	ER <10% and/or PgR <20%	n/a
Residual cancer burden	RCB-III	RCB-0
Genomic signature (Oncotype DX, MammaPrint^®^, EndoPredict^®^, PAM50)	High-risk class	Low-risk class

n/a: not available, RCB: residual cancer burden.

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
