# Peer review of "Definition of High-Risk Early Hormone-Positive HER2−Negative Breast Cancer: A Consensus Review"

_cancers, 2022, doi:10.3390/cancers14081898_

Round 1

Reviewer 1 Report

Well described role of each potential risk factor for recurrence, with very well organized and complete review of literature; however there is not a  specific and detailed description of the additional value of this consensus statement in therapeutic decision

Therefore I suggest in the conclusions a more detailed discussion on the potential addition of the work for clinical daily practice  and management of therapy

Author Response

We thank the reviewer for the constructive suggestion. As suggested, we implemented the discussion/conclusion section of the manuscript.

 The conclusion section of the manuscript now reads: “The IRIDE working group identified an updated list of relapse risk factors for HR+/HER2- eBC (Table 2). The accurate identification and reporting of these features is of pivotal importance in clinical practice to personalize adjuvant treatment intensity. Although each of these factors could modulate the overall risk of relapse, to date a prediction tool that includes and weights each of them is still lacking. However, it appears reasonable to use the identified features to tailor the adjuvant therapy. A possible useful implication of this work resides in a well-characterized list of prognostic items that could represent an essential information list to include in daily clinical practice and clinical trials.

For most of the risk factors considered relevant in our analysis, there was a high agreement rate (Figure 1). Conversely, the agreement for the non-relevant features was generally low, underlying a substantial inconsistency of scientific literature. However, although ctDNA was considered only moderately relevant, a growing prognostic role for this factor might be expected in the near future.

Our comprehensive work produced by an expert consensus represents a pragmatic reference to help clinicians better stratify eBC patients for tailored adjuvant treatments.”

Reviewer 2 Report

Mattia Garutti et al uncovered the definition of high-risk early hormone-positive HER2-negative breast cancer.

The following have not been addressed:

1) The rationale of why the authors came up with this review.

2) What is the information that is not exactly available that motivated the authors to come up with this information. What are the current caveats and how do the authors highlight the current research in answering them? If not they need to address in future directions.

3) HER2-positive breast cancers are aggressive tumors characterized by increased HER2 expression and other genes related to this signaling pathway. Basal-like breast cancer is characterized by aggressive nature, high histological grade, high mitotic index, increased expression of myoepithelial markers, and no expression of ER, progesterone receptor (PR), and HER2. Clinically, they are regarded as TNBC; however, they are not completely synonymous. The normal breast-like group is a poorly defined group that usually does not respond to neoadjuvant chemotherapy. Since they do not express ER, PR, HER2, they might be regarded as triple-negative. However, their genetic profile is not consistent with basal-like breast cancers. TNBC, which accounts for approximately 15% of breast cancer cases, does not overexpress the conventional receptors of breast cancer, i.e., PR, HER2, and ER. Besides the lack of targets for target therapy, the highly aggressive nature of TNBC and the poor prognosis of TNBC patients can justify the need to develop novel therapeutic approaches to treat TNBC patients (please refer to PMID: 34440380 and expand).

4) Does this role of endothelial cells in angiogenesis in a tumor micro-environment involve hypoxia? Since hypoxia is a key factor for angiogenesis, the authors need to substantiate.

5) The authors need to highlight what new information the review is providing to enhance the research in progress.

Author Response

We thank the reviewer for the comments and for the suggestions.

  1. Thank you for this comment. Rational of the present manuscript was reported at line 103-107. As suggested by your comment, we further implemented this rationale section in order to make clearer which recent evolutions of the clinical scenario led to this review. The rationale section now reads: “In this complex and rapidly changing situation, not all clinical scenarios can be directly informed by data from randomized trials. The present consensus panel was established with the aim of identifying clinical, pathological, morphological and genetic factors potentially related to a high risk of recurrence, to better inform treatment decisions in patients with HR+/HER2- early BC (eBC). In the context of the recent increase in potential treatment options for patients with HR+/HER2- eBC and of evidence concerning the use of gene-expression assays in HR+/HER2- eBC, a comprehensive and updated overall view of available data is in fact needed to identify clinically significant recurrence risk factors and better inform day-to-day clinical decision.”

  1. We have expanded these considerations as suggested in both the introduction section of the manuscript and the conclusion section of the manuscript.

  1. We decided to focus our work and consensus on hormone-positive (HR) breast cancers (BCs) excluding triple negative BCs and HER2-positive BCs. This choice reflects the complex decisional process that afflicts clinicians in the adjuvant treatment proposal in the HR+/HER2- BC population, while different considerations are usually applied by the clinician in the context of triple negative or HER2+ BC as highlighted by the reviewer.

  1. As the focus of our review was a clinical one, we decided to exclude preclinical considerations from our work in order to delineate a pragmatical scenario that could aid clinicians in daily practice.

  1. Thank you for this comment. We agree with the reviewer and therefore implemented further the discussion of this point in the conclusion section of the manuscript.

Round 2

Reviewer 2 Report

Is this reviewer feeling that the review would have still benefited from an expansion by providing a perspective (in the form of a short snapshot) of the suggested TNB translational standpoint, but the peer-reviewing is also a subjective topic. The decision based on the previous comments is for the editor besides this, the authors have clarified several of the questions I raised in my previous review. Unfortunately, most of the other major problems have been addressed by this revision.

Author Response

We thank the reviewer for the suggestions and improvements to our manuscript. In particular, we provided more background information concerning the subject of this review and the rationale with drove this consensus.

The IRIDE working group was established with the aim of reviewing the literature evidence to synthesize the current relevant features that predict hormone-positive/HER2-negative early breast cancer relapse. A Panel of experts in breast cancer was involved to identify clinical, pathological, morphological, and genetic factors. A RAND consensus method was used to define the relevance of each risk factor.